# PerFRDiff: Personalised Weight Editing for Multiple Appropriate Facial Reaction Generation

## ABSTRACT

Human facial reactions play crucial roles in dyadic human-human interactions, where individuals (i.e., listeners) with varying cognitive process styles may display different but appropriate facial reactions in response to an identical behaviour expressed by their conversational partners. While several existing facial reaction generation approaches are capable of generating multiple appropriate facial reactions (AFRs) in response to each given human behaviour, they fail to take human's personalised cognitive process in AFRs' generation. In this paper, we propose the first online personalised multiple appropriate facial reaction generation (MAFRG) approach which learns a unique personalised cognitive style from the target human listener's previous facial behaviours and represents it as a set of network weight shifts. These personalised weight shifts are then applied to edit the weights of a pre-trained generic MAFRG model, allowing the obtained personalised model to naturally mimic the target human listener's cognitive process in its reasoning for multiple AFRs generations. Experimental results show that our approach not only largely outperformed all existing approaches in generating more appropriate and diverse generic AFRs, but also serves as the first reliable personalised MAFRG solution. Our code is provided in the Supplementary Material.

## CCS CONCEPTS

• **Human-centered computing** → **Interactive systems and tools**; • **Computing methodologies** → **Computer vision problems**; *Learning latent representations*.

## KEYWORDS

Facial Reaction, Personalisation, Weight Editing

## 1 INTRODUCTION

Human facial reaction plays a crucial role in social interactions, which conveys individuals' (called listeners in this paper) intentions and emotional states in response to behaviours expressed by their conversational partners (called speakers in this paper) [31]. As a result, the capability to promptly generate realistic and appropriate (i.e., contextually suitable) human-style facial reactions in real-time is essential in the development of humanoid virtual agents [46], as it facilitates natural, engaging and empathetic interactions

**Unpublished working draft. Not for distribution.**

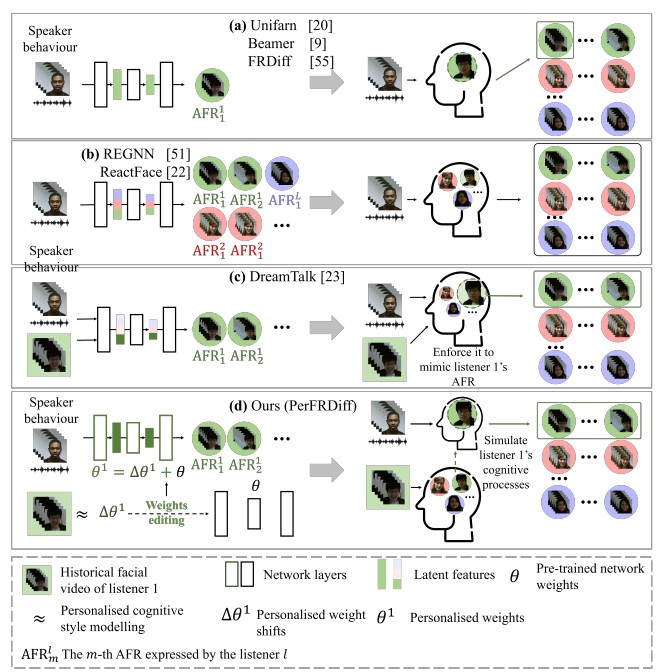

**Figure 1: Comparison between our PerFRDiff with existing approaches: (a) deterministic approaches which aims to reproduce the corresponding 'GT' real appropriate facial reaction (AFR) specifically expressed by a human listener for responding to the input speaker behaviour [8, 17, 52]; (b) distribution-based approaches which aims to generate multiple generic real AFRs from the input speaker behaviour [20, 51]; (c) personalised facial reaction generation approach which generates personalised AFRs by feeding personalised factors to the pre-trained network, enforcing the network to mimic the AFR expressed by the target listener [21]; (d) our approach generates learns a set of unique weights to obtain a personalised network with personalised weights, which naturally simulates personalised cognitive process of the target listener for generating personalised AFRs.**

between human users and digital entities, thereby enhancing the user experience and interaction quality.

Early facial reaction generation approaches [9, 11, 27, 49] frequently attempted to reproduce the paired real facial reaction (i.e., called the 'GT' real facial reaction) triggered by each target speaker behaviour, which treats the facial reaction generation task as a 'one-to-one mapping' problem. However, in real-world dyadic interaction scenarios, the same speaker behaviour can trigger different real facial reactions expressed by different human listeners [22], and thus *facial reaction generation is an actually 'one-to-many mapping' task* [41]. For example, when facing a funny scenario, human

listeners with extroverted personality might react with a broad smile, while listeners with introverted personality might offer only a slight grin. As a result, instead of reproducing the corresponding single GT real facial reaction for each speaker behaviour, Multiple Appropriate Facial Reaction Generation (MAFRG) approaches target at generating multiple diverse facial reactions that are appropriate for responding to each speaker behaviour have been widely developed in the past year [8, 17, 20, 39, 51, 52]. These approaches either learn a distribution [8, 19, 51, 52] or a codebook [17] from each input speaker behaviour, based on which a set of different but appropriate facial reactions (AFRs) can be sampled.

While human listeners' facial reactions in response to their conversational partners are largely depending on their personalised cognitive processes [22, 28, 34, 38, 45], i.e., personalised cognitive processes are shaped by their personalised cognitive styles referring to 'consistent individual differences in ways of organising and processing information and experience' [23, 43], none of existing MAFRG approaches specifically considered this crucial factor in AFRs generation. In other words, personalised AFR generation remains under-explored (**Problem 1**). In this sense, we draw our attention to model personalisation which allows a network to fit on a specific target. Specifically, existing model personalisation approaches are mainly achieved by fine-tuning models on personalised data [2, 13, 13, 15, 26, 32, 34, 38, 50? ] or additionally taking encoded personalised factors as an additional input [5, 12, 33, 35, 48], to produce personalised portrait image generation [26], talking head [21] or lip sync video [15]. Although the latter approaches avoid the complex and time-consuming personalised fine-tuning for every target individual, as illustrated in Fig. 1, they treat the encoded personalised factors as an **external** input to the model for personalised feature refinement, and thus still fail to naturally simulate human cognitive processes which are shaped by **internal** factors (e.g., cognitive style [43] and personality [38]) in human facial reaction generation (**Problem 2**).

In this paper, we propose the first online personalised MAFRG approach (called PerFRDiff) which can generate multiple diverse and appropriate facial reactions in response to each input human audio-visual speaker behaviour, where the personalised cognitive style of the listener is specifically considered. To address the 'one-to-many mapping' problem occurred in MAFRG task, our approach starts with training a diffusion-based Generic Appropriate Facial Reaction Generator (GAFRG), as its denoising process naturally allows multiple different outputs (e.g., facial reactions) to be sampled based on the same condition (e.g., the input speaker behaviour) via randomly picked and different input Gaussian noises [6, 18, 24, 30, 30, 36], i.e., achieving 'one-to-many mapping' network via 'one-to-one training'. The obtained GAFRG aims to simulate generic AFRs in response to each input speaker behaviour, simulating generic/basic cognitive processes commonly shared by different listeners [25]. Then, a Personalised Cognitive Style Modelling (PCSM) module is proposed to model the target listener's personalised cognitive style from the listener's previous facial behaviours (i.e., a short historical face video), which is represented in the form of a set of personalised weight shifts (**addressing Problem 1**). These learned personalised weight shifts are then applied to edit pre-trained weights of the GAFRG, resulting in a personalised instance defined by personalised weights. Manipulating the weights according to the target

listener alters the fundamental way in which the GAFRG perceives the input speaker behaviour as well as how facial reactions in the latent space are formed and transformed throughout the network. In other words, this strategy allows the obtained personalised instance to naturally simulate the corresponding listener's personalised cognitive processes involved in facial reaction generation in the form of its personalised weights (**addressing Problem 2**). The main contributions of this paper are summarised as follows:

- We propose the first online personalised MAFRG model that can simulate personalised cognitive processes of the target listener for generating multiple diverse, personalised, realistic and appropriate facial reactions in response to each speaker behaviour, where diffusion style process is employed to address the 'one-to-many mapping' problem occurred in MAFRG model's training.
- We propose to model the personalised cognitive style of the target listener in the form of personalised weight shifts, and apply them to edit a pre-trained model (GAFRG) which simulates commonly shared cognitive processes among different listeners. This results in a personalised instance particularly representing the target listener's cognitive processes for facial reaction generation.
- Experimental results show that our approach largely outperforms existing state-of-the-art approaches with over twofold appropriateness (FRCorr) improvement in both MAFRG and PMAFRG tasks, and approximately 67%, 188% and 239% times better in terms of diversity over FRDiv, FRDvs and FRVar, respectively.

## 2 RELATED WORK

Early deterministic facial reaction generation approaches [9, 11, 27, 49] were developed to reproduce the GT real facial reaction from each input speaker behaviour. For example, Huang et al. [10] presented a conditional Generative Adversarial Network (GAN) to generate facial reaction sketches conditioned on the facial expressions expressed by the conversational partner, which are expected to match the face sketches extracted from the paired GT real facial reaction. Woo et al. [49] utilised Recurrent Neural Networks (RNNs) to generate a multi-dimensional facial reaction signal (e.g., facial expressions, head motion, posture, etc.) in a real-time dyadic interaction. Song et al. [34, 38] simulated each listener's personalised cognitive processes using a personalised convolutional neural network (CNN) to reproduce the GT real facial reaction.. Recently, several approaches have been proposed to generate multiple diverse but appropriate facial reactions in response to each speaker behaviour. Specifically, Xu et al. [51] and Luo et al. [20] learn a distribution (i.e., Gaussian Mixture Model (GMM) / Gaussian distribution) from each input speaker behaviour, where multiple different AFRs in response to the given speaker behaviour can be sampled from it. This idea has been followed by multiple approaches [8, 52] presented in the REACT2023 challenge [39]. For example, the BEAMER [8] extended the Transformer-based Variational Autoencoder (VAE) architecture provided in [39] to predict a latent AFR distribution for each speaker behaviour. A contrastive loss between the speaker behaviour representation and the predicted facial reaction representation is computed to facilitate the model training. The winner

**Figure 2: The pipeline of the proposed PerFRDiff.** At the time $t$, (1) it starts with a GAFRG that takes a noise sampled from a Gaussian distribution and the given audio-visual speaker behaviour previously expressed at the time $[1, t - w]$ as the input, and generate multiple generic appropriate facial reactions (AFRs) that can be expressed by different human listeners via $D$ steps, where the speaker behaviour semantics are encoded from the speaker behaviour via the SBE module. (2) Then, the PCSM module models the target listener's personalised cognitive style $p^l$ in the form of personalised weight shifts $\Delta\theta^l$ from the listener's historical facial behaviour $F_h^l$. These weight shifts $\Delta\theta^l$ are subsequently applied to define a personalised instance GAFRG$^{\theta^l}$ from the GAFRG, which particularly simulates the target listener $l$'s cognitive processes in facial reaction generation. (3) Finally, the personalised instance GAFRG$^{\theta^l}$ generates a set of personalised AFR segments (each contains $w$ frames) in response to the given speaker behaviour $B_{[1:t-w]}^s$ through a denoising process guided by the its behaviour semantics $\{\bar{E}_{[1:t-w]}^{\text{aud}}, \bar{E}_{[1:t-w]}^{\text{emo}}, \bar{E}_{[1:t-w]}^{\text{app}}\}$.

[17] of the challenge represents a finite set of real facial reactions provided in the training set through a codebook, from which *top-k* AFRs are selected for responding to the input speaker behaviour.

## 3 TASK DEFINITION

In dyadic interaction scenarios, the goal of the Personalised Multiple Appropriate Facial Reaction Generation (PMAFRG) task is to learn a machine learning (ML) model that can generate multiple ($M$) different but appropriate personalised facial reactions $\mathbb{R}^l = \{\hat{R}^l(1), \cdots, \hat{R}^l(M)\}$ in response to the given speaker behaviour $B^s$. This can be formulated as:

$$\mathbb{R}^l = \mathcal{H}^{\text{PMAFRG}}\left(B^s, F_h^l\right) \tag{1}$$

where $F_h^l$ denotes an arbitrary facial behaviour previously expressed by the target listener $l$ (i.e., historical personalised facial behaviour). Here, the generated $\hat{R}^l \in \mathbb{R}^l$ should be similar to at least one real AFR expressed by the target listener $l$ (i.e., a **personalised** real AFR $F^l(n) \in \mathbb{F}^l$) provided in the training set as:

$$\hat{R}^l(m) \approx F^l(n) \in \mathbb{F}^l, \tag{2}$$

where $\mathbb{F}^l$ represents a set of real facial reactions expressed by the target listener $l$ in response to human behaviours that are similar to the given speaker behaviour $B^s$. In particular, the **online PMAFRG model** is expected to predict each personalised AFR

$\hat{R}^l(m)$ in a progressive way that it predicts the current AFR segment $\hat{R}_{[t-w+1:t]}^l(m)$ (i.e., represented by a small video segment consisting of $w$ frames at the time interval $[t - w + 1 : t]$) belonging to $\hat{R}^l(m)$ based on the previous speaker behaviour $B_{[1:t-w]}^s$ expressed at the time $[1 : t - w]$ and a historical facial behaviour $F_h^l$ of the target listener as:

$$\hat{R}_{[t-w+1:t]}^l(m) = \mathcal{H}^{\text{PMAFRG}}\left(B_{[1:t-w]}^s, F_h^l\right) \tag{3}$$

where $w$ simulates the time delay due to the execution of human cognitive processes [4, 38]. The progressively predicted facial reaction segments $\{\hat{R}_{[1:w]}^l(m), \hat{R}_{[w+1:2w]}^l(m), \cdots \hat{R}_{[t-w+1:t]}^l(m)\}$ along the time $[1 : t]$ forms a complete facial reaction $\hat{R}_{[1:t]}^l(m)$ in response to the speaker behaviour $B_{[1:t]}^s$. This task is different from the general MAFRG task defined in [41] and the general online MAFRG task implemented in [20, 41], which only requires each generated AFR to be similar to at least one real AFR that can be expressed by different individuals in response to $B^s$. Please refer to [41] for more details.

## 4 METHODOLOGY

**Overview:** Given an audio-visual speaker behaviour $B_{[1:t-w]}^s = \{A_{[1:t-w]}^s, F_{[1:t-w]}^s\}$ expressed at the time interval $[1 : t - w]$ (i.e., $A_{[1:t-w]}^s$ denotes the audio speaker behaviour and $F_{[1:t-w]}^s$

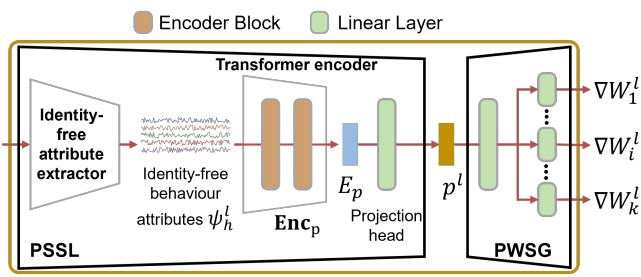

**Figure 3: The architecture of PCSM module consisting of PSSL and PWSG blocks for personalised cognitive style modelling.**

denotes the facial speaker behaviour), and an arbitrary historical facial behaviour segment $F_h^l$ (can have an arbitrary number of frames) previously expressed by the target listener $l$, the proposed PerFRDiff simulates multiple listener $l$'s personalised appropriate facial reactions (AFRs) based on three modules: (i) a **Speaker Behaviour Encoding (SBE)** module which encodes the facial reaction-related audio and facial behavioural semantics from the speaker audio-visual behaviour $B_{[1:t-w]}^s$. Specifically, the SBE encodes three equal-size semantic embeddings from $B_{[1:t-w]}^s$ via three separate encoders $\{\mathbf{Enc}_{\mathrm{aud}}, \mathbf{Enc}_{\mathrm{app}}, \mathrm{and} \mathbf{Enc}_{\mathrm{emo}}\}$, describing speaker audio behavioural semantics $\bar{E}_{[1:t-w]}^{\mathrm{aud}}$, facial emotional semantics $\bar{E}_{[1:t-w]}^{\mathrm{emo}}$, and facial appearance semantics $\bar{E}_{[1:t-w]}^{\mathrm{app}}$ as:

$$
\begin{aligned}
\bar{E}_{[1:t-w]}^{\mathrm{aud}} &= \mathbf{Enc}_{\mathrm{aud}}(E_{[1:t-w]}^{\mathrm{aud}}) \\
\bar{E}_{[1:t-w]}^{\mathrm{emo}} &= \mathbf{Enc}_{\mathrm{app}}(E_{[1:t-w]}^{\mathrm{app}}) \\
\bar{E}_{[1:t-w]}^{\mathrm{app}} &= \mathbf{Enc}_{\mathrm{emo}}(E_{[1:t-w]}^{\mathrm{emo}});
\end{aligned}
\tag{4}
$$

To achieve this, we pre-process the input speaker facial behaviour $F_{[1:t-w]}^s$ into two sets of frame-level facial features: 3D Morphable Model (3DMM) coefficients [47] $E_{[1:t-w]}^{\mathrm{m}}$ representing the facial appearance, as well as facial emotional representations $E_{[1:t-w]}^{\mathrm{e}}$ (i.e., action units (AUs), facial expression probabilities, valence and arousal intensities) [19, 42]. The raw speaker audio signal $A_{[1:t-w]}^s$ is represented by Mel-frequency Cepstral Coefficients (MFCC)) $E_{[1:t-w]}^{\mathrm{a}}$. Here, both $\mathbf{Enc}_{\mathrm{aud}}$ and $\mathbf{Enc}_{\mathrm{app}}$ are a linear layer, while $\mathbf{Enc}_{\mathrm{emo}}$ is a pre-trained RNN-based VAE [1]; (ii) a **Personalised Cognitive Style Modelling (PCSM)** module which learns the target listener's personalised cognitive style $p^l$ based on the historical facial behaviour $F_h^l$ in the form of a set of personalised weight shifts; (iii) finally, a **personalised instance** GAFRG$^{\theta^l}$ obtained by applying the personalised weight shifts to a pre-trained **Generic Appropriate Facial Reaction Generator (GAFRG)** simulates the personalised cognitive processes for the target listener $l$. It promptly generates multiple diverse personalised AFRs $\mathbb{R}_{[t-w+1:t]}^l = \{\hat{R}_{[t-w+1:t]}^l(1), \cdots, \hat{R}_{[t-w+1:t]}^l(m)\}$ in response to the given speaker behaviour at the time $[t-w+1:t]$, conditioned on the encoded speaker behaviour semantics $\{\bar{E}_{[1:t-w]}^{\mathrm{aud}}, \bar{E}_{[1:t-w]}^{\mathrm{emo}}, \bar{E}_{[1:t-w]}^{\mathrm{app}}\}$. The full pipeline is also illustrated in Fig. 2.

## 4.1 Personalised Cognitive Style Modelling

The Personalised Cognitive Style Modelling (PCSM) module models the target listener's personalised cognitive processes to enable the simulation of the target listener's personalised cognitive processes for generating more appropriate and realistic facial reactions. This is achieved by learning a set of personalised weight shifts to represent the target listener's personalised cognitive style $p^l$ that shapes the listener's personalised cognitive processes in facial reaction generation [4, 43]. Specifically, our PCSM module starts with a **Personalised Style Space Learning (PSSL)** block which encodes each target listener's historical facial behaviour $F_h^l$ into a latent space $\mathcal{Z}$ in which the personalised cognitive styles $p^l$ can be modelled as:

$$
p^l = \mathbf{PSSL}(F_h^l).
\tag{5}
$$

Then, a **Personalised Weight Shift Generation (PWSG)** block is introduced to represent $p^l$ in the form of a set of weight shifts $\Delta\theta$ to produce a personalised instance of the GARFG module.

**Personalised Style Space Learning (PSSL):** The PCSM module is built on the hypothesis that the personalised cognitive style $p^l$ of the listener $l$ is consistent across different facial behaviours $\mathbb{F}^l = \{F^l(1), F^l(2), \cdots, F^l(N)\}$ expressed by this listener, but different from the personalised cognitive style of other listeners [23]., i.e., cognitive styles $p^1, \cdots, p^L$ reflected by facial reactions $F^1(n), \cdots, F^L(n)$ expressed by different listeners for responding to the same speaker behaviour $B^s(n)$ should be different. This can be defined as:

$$
\begin{aligned}
p^l(F^l(1)) &\approx p^l(F^l(2)) \approx \cdots \approx p^l(F^l(N)), \forall l \in \{1, 2, \cdots, L\} \\
p^1(F^1(n)) &\neq p^2(F^2(n)) \neq \cdots \neq p^L(F^L(n)), \forall n \in \{1, 2, \cdots, N\},
\end{aligned}
\tag{6}
$$

where $p^l(F^l(n))$ denotes the cognitive style reflected by the facial reaction $F^l(n)$. To obtain this personalised cognitive style from a target listener's historical facial behaviour video $F_h^l$, our PSSL is learned to map all facial behaviours to a latent space $\mathcal{Z}$, where the latent representations extracted from facial behaviours expressed by the same listener are pulled together while latent representations extracted from facial behaviours belonging to different listeners are pushed apart, i.e., latent representations in $\mathcal{Z}$ should meet Eqa. 6. More details for training our PSSL block are explained in Sec. 4.3. To avoid this process being influenced by identities of listeners, which are also consistent in different face videos of the same listener but different across various listeners, our PSSL first extracts a set of identity-free facial behaviour attributes $\psi_h^l$ (e.g. 3DMM and Action Units (AUs)) from each frame of the given historical face video $F_h^l$ (as shown in Fig. 3), aiming to model the personalised cognitive style underlying the facial behaviours regardless of the listener's identity. Then, the $\psi_h^l$ is encoded as an embedding $E_p$ through a Transformer encoder $\mathbf{Enc}_p$ [44] to capture long-range dependencies within the given sequential behaviour, as the target personalised cognitive style $p^l$ should be consistent across the entire video $F_h^l$ [23]. Finally, the obtained $E_p$ is projected into the target latent space $\mathcal{Z}$ through a projection head (i.e., a linear layer) as the required $p^l$.

**Personalised Weight Shifts Generation (PWSG):** Based on the personalised cognitive style $p^l$, the PWSG block simulates the corresponding listener's cognitive processes involved in the facial reaction generation by learning a set of unique and personalised

weight shifts $\Delta\theta^l$ from $p^l$ via a multi-branch network as:

$$\Delta\theta^l = \mathbf{PWSG}(p^l). \tag{7}$$

These personalised weight shifts are then applied to transform the pre-trained generic appropriate facial reaction generator (GAFRG) as a personalised instance (i.e., a personalised appropriate facial reaction generator) $\mathbf{GAFRG}^{\theta^l}$ representing the personalised facial reaction cognitive process of the listener $l$. Here, each branch of the PWSG network generates a set of weight shifts $\Delta W_k^l$ to personalise the $k_{\text{th}}$ layer $\text{GAFRG}_k$ in the GAFRG.

## 4.2 Personalised Appropriate Facial Reaction Generation

The PAFRG module aims to naturally simulate the personalised cognitive process of the target listener in MAFRG task without requiring time-consuming personalised fine-tuning. Specifically, it first pre-trains a generic Appropriate Facial Reaction Generator (GAFRG) that is capable of generating multiple generic AFRs in response to each input speaker behaviour, simulating generic cognitive processes commonly shared by different listeners. Then, the obtained $p^l$ representing the target listener $l$'s personalised cognitive style is applied to edit the GAFRG's weights, producing a personalised instance $\mathbf{GAFRG}^{\theta^l}$, which particularly simulates the personalised cognitive processes of the listener $l$ in this listener's facial reaction generation.

**Generic Appropriate Facial Reaction Generator:** Separately training/fine-tuning a personalised facial reaction generator for each listener is challenging due to the need for not only a large amount of interaction data expressed by every target listener but also substantial computational costs. Since there are commonly shared facial reactions that can be expressed by different human listeners in response to the same speaker behaviour (i.e., some facial reaction styles are commonly shared by various human listeners), we first learn a GAFRG aiming to generate generic AFRs that can be expressed by different listeners, simulating the commonly shared generic cognitive processes involved in facial reaction generation. To address the 'one-to-many mapping' problem occurring in MAFRG models' training, the employed GAFRG is designed as a transformer-based diffusion model [44] which reformulates the 'one-to-many mapping' problem as a 'one-to-one mapping' task by denoising each noisy real AFR (ground-truth provided in the training set) degraded by a random Gaussian noise to itself (a real AFR) expressed by a real listener in response to the speaker behaviour $B_{[1:t-w]}^s$ during training. This process comprises two main steps: (i) a forward diffusion process where small portions of Gaussian noise are progressively added to a real AFR segment $F_{([t-w+1:t],0)}$ in $D$ steps, resulting in a noisy AFR segment $F_{([t-w+1:t],D)}$ at the final step $D$; and (ii) a reverse (denoising) process where the GAFRG denoises $F_{([t-w+1:t],D)}$ step-by-step to recover the original AFR segment $F_{([t-w+1:t],0)}$ expressed at the time $[t-w+1:t]$. At each reverse step, given the speaker behaviour $B_{[1:t-w]}^s$ and a noisy version of AFR segment $F_{([t-w+1:t],d)}$ obtained at the $d$-th diffusion step, the GAFRG learns to predict the added noise $\epsilon_\theta(F_{([t-w+1:t],d)}, B_{[1:t-w]}^s, d)$ at the $d$-th diffusion step and removes it from $F_{([t-w+1:t],d)}$, reversely obtaining

$F_{t-w+1:t,d-1}$. The reverse process is guided by the input speaker behaviour $B_{[1:t-w]}^s$ expressed at the time $[1:t-w]$ as the condition (i.e., as the key and value), according to which the latent representation of the predicted original AFRs (i.e., as the query) is modified via cross-attention operations. *This process encourages the GAFRG to understand the relationships between the speaker behaviour $B_{[1:t-w]}^s$ and its corresponding multiple AFRs $\mathbb{F}_{[t-w+1:t]} = \{F_{[t-w+1:t]}(1), F_{[t-w+1:t]}(2), \cdots, F_{[t-w+1:t]}(N)\}$ (i.e., the distribution of them).* As a result, the well-trained GAFRG can generate multiple AFRs $\mathbb{R}_{t-w+1,t}$ from randomly sampled Gaussian noises conditioned on the input speaker behaviour $B_{[1:t-w]}^s$ at the inference stage.

**Simulating personalised cognitive process for AFRs generation:** To consider personalised cognitive style of the target listener $l$ in generating multiple AFRs $\mathbb{R}^l = \{\hat{R}_{[1:t]}(1), \cdots, \hat{R}_{[1:t]}(m)\}$ in response to the speaker behaviour $B_{[1:t]}^s$, we apply the personalised weight shifts $\Delta\theta$ learned from the target listener $l$'s personalised cognitive style $p^l$, to edit the weights of the obtained GAFRG $\theta$. This results in a personalised instance of GAFRG, denoted as $\text{GAFRG}^{\theta^l}$, with its personalised weights fundamentally determining the way that $\text{GAFRG}^{\theta^l}$ interprets the input speaker behaviours and forms a set of AFRs throughout the network. Let $\theta$ be the weights of the pre-trained GAFRG, this process can be formulated as:

$$\theta^l = \theta + \Delta\theta^l, \tag{8}$$

where $\theta^l$ denotes the weights of a personalised instance of GAFRG module representing the listener $l$'s cognitive process, and $\Delta\theta^l$ denotes the personalised weight shifts modelled by the PCSM module. Consequently, each personalised facial reaction segments $\mathbb{R}_{[t-w+1:t]}^l = \{\hat{R}_{[t-w+1:t]}^l(1), \cdots, \hat{R}_{[t-w+1:t]}^l(m)\}$ expressed at the time $t$ can be predicted by:

$$\mathbb{R}_{[t-w+1:t]}^l = \mathbf{GAFRG}^{\theta^l}(B_{[1:t-w]}^s, \mathbb{R}_{[1:t-w]}^l), \tag{9}$$

where $\mathbb{R}_{[1:t-w]}^l$ denotes the previously predicted facial reaction sequence at the time $[1:t-w]$. As a result, consecutively predicted personalised facial reaction segments $\{\hat{R}_{[1:w]}^l(m), \hat{R}_{[w+1:2w]}^l(m), \cdots, \hat{R}_{[T-w+1:T]}^l($ finally form a complete facial reaction $\hat{R}_{[1:T]}^l(m)$ consisting of $T$ frames in response to a speaker behaviour $B_{[1:T]}^s$. This way, multiple personalised AFRs in response to a speaker behaviour $B_{[1:T]}^s$ can be obtained by repeating the consecutive prediction process.

## 4.3 Training Strategy and Loss Functions

Our PerFRDiff is trained via a two-stage strategy, where the first stage includes GAFRG and PSSL's training, while the second stage focuses on training the PWSG block.

**Pre-training GAFRG:** The GAFRG is jointly trained with the SBE module by performing a reverse diffusion process where the GAFRG denoises a noisy facial reaction $F_{([t-w+1:t],d)}$ to recover the original clean real AFR $F_{([t-w+1:t],0)}$ triggered by the speaker behaviour $B_{[1:t-w]}^s$ expressed at the time $[1:t-w]$, which is supervised by optimising an MSE loss as:

$$\begin{aligned}\mathcal{L}_1 =\mathbb{E}_{\mathbf{F}_{([t-w+1:t],0)},\epsilon}\big[\|F_{([t-w+1:t],0)}- \\ \hat{F}_{([t-w+1:t],0)}(F_{([t-w+1:t],d)}, c, d)\|^2\big],\end{aligned} \tag{10}$$

where $\hat{F}_{([t-w+1:t],0)}(F_{([t-w+1:t],d)}, c, d)$ denotes the predicted original clean AFR based on the $F_{([t-w+1:t],d)}$ in the reverse process, and $c$ denotes the conditions including the speaker behaviour $B^s_{[1:t-w]}$ and previously predicted facial reaction $\mathbb{R}^l_{[1:t-w]}$.

**Training PSSL:** Let $\mathbb{F}^l$ be a set of real AFRs expressed by the $l$-th listener in response to the speaker behaviour $B^s$; $\mathbb{F} = \{\mathbb{F}^1 \cup \mathbb{F}^2 \cup \cdots \cup \mathbb{F}^l\}$ be a union set of $X$ real AFRs expressed by $L$ different listeners in response to $B^s$; $i \in I \equiv \{1 \ldots X\}$ be the index of a real AFR in $\mathbb{F}$; $G(i) \equiv I \backslash \{i\}$ be the set of indices excluding $i$; and $Q(i)$ denotes the set of indices of AFRs belonging to the same listener as the $i$-th AFR and $i \notin Q(i)$. The training process of the PSSL block is achieved by minimising a contrastive loss [14] as:

$$\mathcal{L}_2 = \sum_{i \in I} \frac{-1}{|Q(i)|} \sum_{q \in Q(i)} \log \frac{\exp(p(i) \cdot p(q)/\tau)}{\sum_{g \in G(i)} \exp(p(i) \cdot p(g)/\tau)}, \quad (11)$$

where $p(i)$ denotes the personalised cognitive style derived from the $i$-th AFR in $\mathbb{F}$, $|Q(i)|$ denotes the cardinality of $Q(i)$, and $\tau$ is a temperature parameter. This loss maximises the similarity between each pair of the personalised cognitive styles $(p(i), p(q))$ modelled from real AFRs belonging to the same listener.

**Training PWSG:** In this stage, we integrate the PSSL and PWSG blocks into the GAFRG, then freeze the well-trained PSSL block and GAFRG. Subsequently, we employ the loss defined in Equation 10 to optimise the PWSG block. Let $W_k \in \theta$ be the weight matrix of the $k$-th layer GAFRG$_k$ to be edited in the GAFRG and $\Delta W^l_k$ be the personalised weight shifts generated by the PWSG block. The forward propagation of the updated layer GAFRG$_k^{\theta^l}$ is:

$$z^l_k = \sigma\left((W_k + \Delta W^l_k)z^l_{k-1} + b_k\right), \quad (12)$$

where $z^l_k$ denotes the output of the target layer GAFRG$_k^{\theta^l}$; $b_k$ denotes the biases at the target layer GAFRG$_k^{\theta^l}$, and $\sigma$ is an activation function. Correspondingly, the gradient of the loss $\mathcal{L}_1$ with respect to the personalised weight shift $\Delta W^l_k$ generated by the PWSG block is computed using the rule chain as:

$$\frac{\partial \mathcal{L}_1}{\partial(\Delta W^l_k)} = \frac{\partial \mathcal{L}_1}{\partial z^l_k} \cdot \frac{\partial z^l_k}{\partial((W_k + \Delta W^l_k)z^l_{k-1} + b_k)}$$
$$\cdot \frac{\partial((W_k + \Delta W^l_k)z^l_{k-1} + b_k)}{\partial(\Delta W^l_k)} = (z^l_{k-1})^T \cdot \left(\frac{\partial \mathcal{L}_1}{\partial z^l_k} \cdot \sigma'\right), \quad (13)$$

where $(z^l_{k-1})^T$ denotes the transpose of $z^l_{k-1}$, and $\sigma'$ denote the derivative of the activation function $\sigma$. The gradient information $\frac{\partial \mathcal{L}_1}{\partial(\Delta W^l_k)}$ can be further back-propagated through the PWSG block for the purpose of training. It is worth noting that the weights of the GAFRG are only updated based on the weight shifts generated by the PWSG block in the forward propagation at this stage, rather than being optimised in the backpropagation.

# 5 EXPERIMENTS

## 5.1 Experimental Settings

**Dataset:** Our PerFRDiff is evaluated on a publicly accessible audio-visual dyadic interaction dataset provided by REACT2024 challenge [40] [1]. It contains 2962 pairs of human speaker-listener dyadic interaction clips (including 1593 training pairs, 562 validation pairs and 806 test pairs) recorded under various contexts, with each clip lasting for 30 seconds. These clips are originally recorded by two datasets: NoXI [3] and RECOLA [29].

**Implementation Details:** We first crop the face region from every frame of the given clips and resize it into $224 \times 224$. Then, our PerFRDiff is trained with the Adam [16] optimizer with the initial learning rate of 0.0001, coupled with a cosine-annealing schedule, for optimizing GAFRG and PSSL. Additionally, we employ the SGD optimizer with $lr = 0.001$ to train our PWSG. Our diffusion-based GAFRG benefits from the classifier-free guidance [7] which enhances the balance between the quality and diversity of the generated facial reactions. The sampling process is implemented using a DDIM sampler [37]. All experiments are conducted on Nvidia A100 GPUs using PyTorch. More details about implementation and datasets are available in the Supplementary Material.

**Metrics:** Following previous challenges [39, 40], we evaluate four aspects of the generated AFRs, including: **appropriateness** (measured by FRCorr and FRdist), **diversity** (i.e., FRDiv, FRDvs and FRVar, where FRDiv measures the diversity of multiple generated AFRs in response to the same speaker behaviour), **realism** (FRRea) and **synchrony** (FRSyn). Please refer to [41] for more details about these metrics.

## 5.2 Comparison with existing approaches

Table 1 compares our PerFRDiff with existing MAFRG approaches on both MAFRG task (i.e., requiring each generated facial reaction to be similar to at least one real AFRs expressed by human listeners) and PMAFRG task (i.e., requiring each generated facial reaction to be similar to at least one real AFRs expressed by the target listener (the one who responded to the input speaker behaviour in the training set)). It can be observed that our PerFRDiff demonstrates large advantages over all existing approaches in generating both generic AFRs (i.e., MAFRG task) as well as personalised AFRs (i.e., PMAFRG task), which are evidenced by the highest FRCorr results (0.36 and 0.38), i.e., it brings 100% and 110% FRCorr improvements over previous state-of-the-art approaches (i.e., REGNN and Unifarn) on the MAFRG and PMAFRG tasks. Compared to other approaches, the PerFRDiff also shows significant improvements on all three diversity and realism metrics: FRDiv, FRDvs, and FRVar (more than 67%, 188%, and 239%, respectively). This suggests that our PerFRDiff **is capable of generating diverse but appropriate and realistic facial reactions in response to not only the same speaker behaviour (indicated by the highest FRDiv) but also different speaker behaviours (indicated by the highest FRDvs)**. It should be noted that the reason behind consistent superior appropriateness performance achieved by the same systems on the MAFRG task over the PMAFRG task is that given an input speaker behaviour, its corresponding generic real AFRs (labels for MAFRG task) include not only its personalised real AFRs (labels for PMAFRG task) but also real AFRs expressed by other listeners. Fig. 4 further visualizes the AFRs generated by our approach and competitors.

---

[1]https://sites.google.com/cam.ac.uk/react2024/home

**Table 1: Quantitative comparison between our PerFRDiff and existing MAFRG methods on REACT 2024 test set for both MAFRG and PMAFRG tasks. The best result achieved for each metric is marked in bold.**

| Method | Appropriateness | | | | Diversity ($\times 10^{-2}$) | | | Realism | Synchrony |
|---|---|---|---|---|---|---|---|---|---|
| | FRCorr ↑ | | FRdist ↓ | | FRDiv ↑ | FRDvs ↑ | FRVar ↑ | FRRea ↓ | FRSyn ↓ |
| | MAFRG | PMAFRG | MAFRG | PMAFRG | | | | | |
| Chance level | 0.05 | 0.01 | 237.21 | 194.02 | 16.67 | 8.33 | 16.67 | - | 43.84 |
| Trans-VAE [39] | 0.09 | 0.08 | 98.31 | 105.27 | 3.04 | 3.16 | 0.37 | 67.74 | 44.86 |
| REGNN [51] | 0.19 | 0.17 | **84.54** | **91.33** | 0.07 | 3.42 | 0.61 | - | **41.35** |
| Unifarn [17] | 0.19 | 0.15 | 98.51 | 103.21 | 8.19 | 7.60 | 2.59 | - | 46.11 |
| Beamer [8] | 0.11 | 0.10 | 97.33 | 105.09 | 5.08 | 3.74 | 1.96 | - | 48.12 |
| FRDiff [52] | 0.14 | 0.13 | 91.05 | 98.28 | 6.90 | 4.43 | 1.08 | 69.37 | 47.66 |
| BeLFusion [1] | 0.12 | 0.11 | 94.16 | 98.95 | 3.60 | 3.84 | 2.49 | 78.96 | 49.00 |
| PerFRDiff (Ours) | **0.38** | **0.36** | 94.72 | 98.43 | **13.68** | **21.91** | **8.79** | 47.62 | 45.28 |

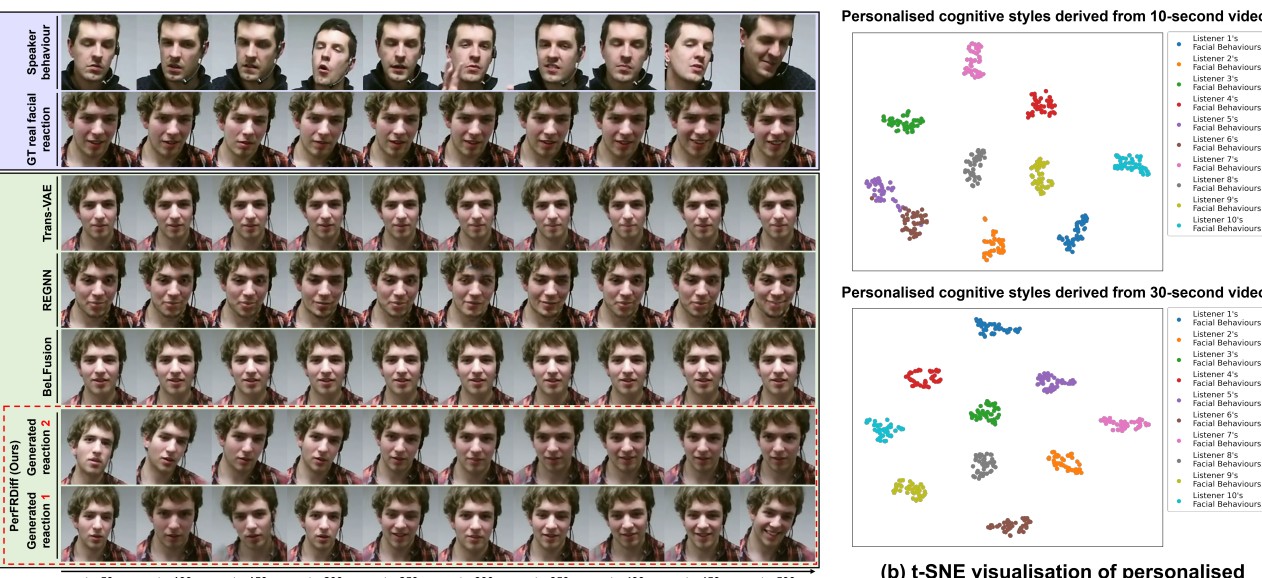

**Personalised cognitive styles derived from 10-second videos**

**Personalised cognitive styles derived from 30-second videos**

$t=50$   $t=100$   $t=150$   $t=200$   $t=250$   $t=300$   $t=350$   $t=400$   $t=450$   $t=500$

**(a) Visualisation of generated facial reactions**

**(b) t-SNE visualisation of personalised cognitive styles derived from 10-second and 30-second listener videos**

**Figure 4: (a) Visualisation of facial reactions generated by different approaches, where ours clearly show more head movements and diverse facial expressions in response to the speaker behaviour; (b) t-SNE visualisation of personalised cognitive styles modelled from varying lengths/contents of listeners' historical facial videos by our PSSL, where personalised cognitive styles modelled from facial videos of varying lengths/contents belonging to the same listener are well clustered together, but separated from other listeners' personalised cognitive styles.**

## 5.3 Ablation Studies

This section conducts a series of ablation studies to evaluate various aspects our PerFRDiff. Additional analysis of three types of inputs, statistical difference analysis, model complexity analysis, and other analysis are provided in the Supplementary Material.

**Contributions of different modalities:** Table 2 evaluates the importance of audio and visual behaviours expressed by the in generating AFRs. It can be observed that using both audio and visual modalities facilitates more appropriate and diverse AFRs generation, as indicated by the highest FRCorr and lowest FRdist. While both

**Table 2: Ablation study results on different modalities.**

| Modality | | FRCorr ↑ | | FRdist ↓ | | FRDiv ↑ |
|---|---|---|---|---|---|---|
| Audio | Visual | MAFRG | PMAFRG | MAFRG | PMAFRG | |
| ✓ | | 0.32 | 0.28 | 171.47 | 174.71 | 0.01 |
| | ✓ | 0.32 | 0.27 | **101.13** | 109.87 | 16.31 |
| ✓ | ✓ | **0.35** | **0.31** | 105.57 | **107.46** | **17.14** |

**Table 3: Results achieved for different GAFRG settings, where 'Input $p^l$' denotes feeding personalised representation to the GAFRG network (Solution 3 in Fig. 1).**

| Paradigm | FRCorr ↑ | | FRdist ↓ | |
|---|---|---|---|---|
| | MAFRG | PMAFRG | MAFRG | PMAFRG |
| GAFRG | 0.35 | 0.31 | 105.57 | 107.46 |
| GAFRG (Input $p^l$) | 0.35 | 0.31 | 100.59 | 103.27 |
| GAFRG$^{\theta^l}$ (Weight editing) | **0.38** | **0.36** | **94.72** | **98.43** |

**Table 4: Results of weight rewriting for different layers.**

| Components | | | | FRCorr (×10$^{-2}$) ↑ | | FRdist ↓ | |
|---|---|---|---|---|---|---|---|
| Self-attn | Cross-attn | Feed-forward | Mapping | MAFRG | PMAFRG | MAFRG | PMAFRG |
| ✓ | | | | 33.56 | 31.90 | 106.40 | 110.92 |
| | ✓ | | | 35.79 | 33.20 | 99.04 | 101.59 |
| | | ✓ | | 35.16 | 32.50 | 104.62 | 107.19 |
| | | | ✓ | 37.16 | 33.52 | 95.05 | 99.98 |
| ✓ | | | ✓ | 36.01 | 33.35 | 100.22 | 103.07 |
| | | ✓ | ✓ | 35.82 | 33.12 | 95.97 | 99.51 |
| | ✓ | | ✓ | **38.45** | **35.84** | **94.72** | 98.43 |
| ✓ | ✓ | ✓ | ✓ | 38.01 | 35.35 | 94.49 | **97.16** |

speaker audio and facial behaviours are important for generating AFRs, excluding speaker facial behaviour results in a larger DTW distance between generated facial reactions and real AFRs as well as very low diversity among facial reactions that are generated to respond to the speaker behaviour. Thus, we conclude that: (1) facial behaviours are more reliable for predict AFRs; (2) while non-verbal audio behaviour is also informative for AFRs generation (i.e., decent FRCorr performance), it cannot trigger diverse facial reactions; (3) speaker audio behaviour provide complementary cues and more details to speaker facial behaviour for AFRs generation, and thus multi-modalities enables our PerFRDiff to generate more appropriate facial reactions in response to it.

**Effectiveness of personalised cognitive style modelling:** Table 3 shows that the best appropriateness results of the generated facial reactions on both tasks are achieved by the personalised instance GAFRG$^{\theta^l}$ achieved by our approach, which naturally simulates the personalised cognitive processes (through weight editing) of the target listener involved in facial reaction generation. This validates our assumption that additionally considering individual differences (i.e., cognitive styles and cognitive processes) is crucial in generating more appropriate facial reactions (high correlations and fewer distances with corresponding real AFRs). More importantly, a common strategy which treats the personalised factors as an external input, i.e., directly feeding the obtained personalised style representation to the MAFRG (Solution 3 in Fig. 1), shows much less effectiveness than our strategy, which further validate our novel cognitive process simulation strategy is suitable for MAFRG and PMAFRG tasks.

**Weight editing of different layers:** Table 4 reports the influences caused by editing the weight matrices of different layers within the GAFRG$^{\theta^l}$ on the MAFRG and PMAFRG tasks. Specifically, the GAFRG layers to be edited include the self-attention (self-attn) block, cross-attention (cross-attn) block, feed-forward block (within the transformer decoder) and the mapping layer (Fig.

**Table 5: Results achieved for varying video lengths.**

| Length | FRCorr ↑ | | FRdist ↓ | | FRDiv ↑ |
|---|---|---|---|---|---|
| | MAFRG | PMAFRG | MAFRG | PMAFRG | |
| 10s | **0.39** | **0.36** | 97.72 | 99.95 | **13.76** |
| 20s | 0.38 | 0.35 | 95.33 | **98.04** | 13.43 |
| 30s | 0.38 | **0.36** | **94.72** | 98.43 | 13.68 |

**Table 6: Results achieved for different video contents.**

| Video | FRCorr (×10$^{-2}$) ↑ | | FRdist ↓ | | FRDiv ↑ |
|---|---|---|---|---|---|
| | MAFRG | PMAFRG | MAFRG | PMAFRG | |
| Video 1 | 38.45 | 35.84 | **94.72** | **98.43** | 13.68 |
| Video 2 | 38.49 | 35.93 | 95.08 | 98.52 | 13.67 |
| Video 3 | **38.65** | **35.99** | 95.05 | 98.51 | 13.68 |
| Video 4 | 38.53 | 35.92 | 95.06 | 98.55 | 13.68 |

2). Our observations are presented as follows: (i) editing the cross-attention block is crucial as it directly interprets the input speaker behaviour, and thus decides the initial understanding of the condition; and (ii) editing multiple components generally enables more appropriate personalised facial reaction generation, compared to editing a single component. We assume this as human facial reaction generation requires multiple steps where there could be more than one step that is person-dependent (not commonly shared by different listeners), and thus personalising/editing multiple components could better simulate such processes of the target listener.

**Influences of the historical facial videos' lengths/contents:** Table 5 first demonstrates that the performances achieved for three different length settings of the historical facial video are similar (with FRCorr around 0.36, FRdist around 98.81, and FRDiv around 13.6), suggesting that our PCSM module can effectively model the personalised cognitive style of the target human listener based on a relatively short facial behaviour video (e.g., 10s). As demonstrated in Table 6 and TSNE figures in Fig. 4, these modelled personalised cognitive styles are also relatively invariant and robust to behaviours expressed by the target listener, regardless of the contents of the expressed facial behaviours, i.e., the personal cognitive styles learned by our approach well meet the rules described in Eqa. 6.

## 6 CONCLUSION

This paper proposes the first online personalised MAFRG approach that can naturally simulate human listener's personalised cognitive processes in the form of a diffusion-style network with personalised weights. Results demonstrate that our approach achieved substantial improvements in both the appropriateness and diversity of the generated facial reactions compared to previous state-of-the-art approaches. Our strategy can also effectively and robustly model personalised cognitive style from an arbitrary historical facial video expressed by the target listener, based on which more appropriate facial reactions can be generated. The key limitation is that the relatively high complexity of our model when a large number of weights need to be edited, which will be addressed in future work.

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
