# OpenReview forum: "PerFRDiff: Personalised Weight Editing for Multiple Appropriate Facial Reaction Generation"
_acmmm.org/ACMMM/2024/Conference — MM2024 Poster_

### Official Review · Reviewer_VmLg · 2024-05-24

**Rating:** 4
**Confidence:** 3

**Summary:**

This paper proposes a personalized multiple facial reaction generation strategy via network weight shift.

**Strengths:**

The personalized weight shift module in proposed PerFRDiff framework is novel. Given any speakers' context and speaking information, the listener reacts according to their personality and cognitive processing of the situation. Thus, the weight editing accordingly is realistic approach.

**Limitations:**

1. From the paper, it is not clear what does it mean by cognitive style. Given a very short interaction video, what are the verbal and nonverbal cues that can be used for representing 'cognitive style'.
2. Also, how to get the facial behavioural history over a short interaction video is not clear.
3. If the personalized cognitive style is derived from 10/30 sec video, how does it relate with multiple reaction generation. Each target listener has one cognitive style.
4. Also, given any context there could be multiple possible reactions (seems like open ended problem). Then, how it is relatable with ground truth for training and evaluation. Please clarify this aspect.
5. Minor issue: the opening quote is in opposite direction in few places, please edit this typo. Also, there is a question mark in line 140.

**Suitability:**

3

---

### Official Review · Reviewer_aSKv · 2024-05-25

**Rating:** 5
**Confidence:** 2

**Summary:**

In this paper, the first online personalised multiple appropriate facial reaction generation (MAFRG) approach is proposed which learns a unique personalised cognitive style from the target human listener’s previous facial behaviours and represents it as a set of network weight shifts. These personalised weight shifts are then applied to edit the weights of a pre-trained generic MAFRG model, allowing the obtained personalised model to naturally mimic the target human listener’s cognitive process in its reasoning for multiple AFRs generations. The authors achieve state-of-the-art results, showing the effectiveness of their method.

**Strengths:**

- The proposed approach that uses personalised weight shifts to edit the weights of a pre-trained generic MAFRG model is novel.
- The authors achieved state of the art results on several aspects of the generated AFRs.
- The paper is well-written in an understandable manner.
- The visual representation is good and explains the proposed methodology quite well.

**Limitations:**

- What is the use case of such a model that generates personalised facial reactions?
- How much is the computational time? Is the method feasible in real-time?
- There are minor typing errors. The content in line 560 is cut.

**Suitability:**

3

---

### Official Review · Reviewer_iyYS · 2024-05-25

**Rating:** 6
**Confidence:** 4

**Summary:**

The paper presents a novel facial reaction generation approach that takes into account the human’s (specifically the listener's ) personalised cognitive process in appropriate facial reaction (AFR) generation. This approach learns a unique personalised cognitive style from the target human listener’s previous facial behaviours which is used in AFR generation.

**Strengths:**

- The problem addressed here is novel, interesting and relevant to the scope of the conference. It is interesting and critical to incorporate the listener's personal cognitive process while the facial reactions are generated.
- The work is well presented and problem statement is well motivated.
- The Experiments are sufficient and the results discussion is interestingly presented

**Limitations:**

-There are too many notations. Defining the notations in a table might be useful for the readers to follow the explanations.
-What does 'GT' stand for in 'GT real facial reaction'
-In addition to quantitative assessment, incorporating user evaluation to assess the diversity and appropriateness of facial reaction generation would be beneficial.
- The facial reaction segments in line 570 is incomplete
- The manuscript should be proofread to correct grammatical errors and typos. Some are given below
-Typo line no 140 - [2, 13, 13, 15, 26, 32, 34, 38, 50? ]
- Typo in reaction.. 218

**Suitability:**

3

---

### Official Review · Reviewer_4ufy · 2024-05-26

**Rating:** 5
**Confidence:** 3

**Summary:**

The paper proposes a method for listeners' facial reaction generation in dyadic interactions. The method learns online the personalised cognitive style of the target listener. It incorporates that information into a pre-trained generic multiple-appropriate facial reaction generation model to mimic the target listener's behaviour. The paper reports significant improvements in this task using the REACT2024 challenge dataset.

**Strengths:**

- The paper proposes an online personalisation approach which leads to significant improvements
- The paper provides a thorough technical description of the proposed methodology
- The paper provides extensive evaluation and discussion of the obtained results

**Limitations:**

- I found the notation used in the paper difficult to follow and over-complicated. For the sake of readability, the paper could benefit from simplified and intuitive notations which will enable smoother reading and comprehension experience.

- Please check the formatting as in several places the text is not well formatted.

- In the provided demo video, it seems that the generated behaviours also involve considerable camera rotations and transloations. This is not observed in the other methods presented. It would be important to discuss why the output is so much influenced by the 'camera rotation' (which I guess happens because of the image cropping) and how this can be mitigated.

**Suitability:**

3

---

### Meta-Review · Area_Chair_oD6p · 2024-07-03

**Recommendation:** Accept (Poster)
**Confidence:** 5

**Metareview:**

2xWA, 1xBA and 1xA ratings
The reviewers appreciate the novelty of the proposed appropriate reaction generation technique. The rebuttal seems to address the queries.